# Spin caloritronics in a hybrid nanomagnet

Long Bai, Furong Tang, and Rong Zhang[*]

*School of Materials Science and Physics,*

*China University of Mining and Technology, Xuzhou 221116, China*

(Dated: December 31, 2024)

## Abstract

Spin caloritronics focuses on the interplay of charge, heat and spins. Herein, we propose a spin-related thermoelectric device based on a hybrid nanomagnet in order to explore physical mechanisms and potential applications for heat-spin-charge interconversion. The evolution characteristics of thermoelectric coefficients in physically allowed parameter regions are investigated at length. Interestingly, not only is the Wiedemann-Franz strongly violated, but also a pure spin-Seebeck effect can be generated, which can be utilized to engineer a pure spin-current generator. Additionally, a considerable spin thermoelectric efficiency can be achieved, and the direction of thermally-induced spin currents can be manipulated, which provides valuable information about thermoelectric characteristics mediated by electron spins. Our results offer a deepgoing insight into spin caloritronc properties of hybrid thermoelectric devices based on nanomagnets.

---
[*]1979zhangrong@163.com

## I. INTRODUCTION

Hybrid quantum dot(QD) systems with superconducting(SC) and metallic reservoirs connected by quantum wires have attracted a lot of interest [1], which stems from the local or nonlocal Andreev reflection(AR) processes in hybrid QD nanostructures. For the local AR(also known as direct AR), when an electron from a metallic electrode is incident on a SC one, it is reflected as a hole coming back to the metal electrode, while a Cooper pair is transmitted into the SC side. The nonlocal AR(also called crossed AR) is different from the local AR, where two electrons with opposite spins from spatially separated metallic electrodes form a Cooper pair. Interestingly, hybrid QD systems offers the unique platforms to investigate the interplay of the AR mechanism and typical phenomena like the Fano effect [2, 3], the Dicke effect [4, 5], the Coulomb blockade [6–8], and the Kondo effect [9, 10]. Furthermore, the pairs of entangled electrons based on the Cooper pair splitting can be created in hybrid QD systems[11, 12], which is very appealing for potential applications in superconducting spintronics, quantum information, and quantum computation.

During the last two decades, the thermoelectric effect or the Seebeck effect, which relates the potential difference $\Delta V$ to the temperature difference $\Delta T$, has attracted enormous theoretical and experimental attention from the perspective of energy harvesting. In general, the thermoelectric conversion efficiency is determined by the dimensionless thermoelectric figure of merit $ZT = S^2GT/k$, where $S$ is the Seebeck coefficient(thermopower), $G$ is the electrical conductance, $T$ is the absolute temperature, and $\kappa$ is the thermal conductance. Due to the limitation of the Wiedemann-Franz(WF) law, $ZT$ values in bulk materials do not exceed one, $ZT \leq 1$. As a consequence, the searching for new materials or systems with the high $ZT$ has constituted the central task of thermoelectrics. It is relieved that the impressive advances in nanotechnology provide the possibility of achieving the significant enhancement of $ZT$ in nanoscale systems [13–15], such as in quantum dots [16–19], molecular junctions [20, 21], quantum wires [22, 23], Graphene nanoribbons [24, 25], and topological materials [26–30].

From an application point of view, hybrid QD systems can be utilized as thermoelectric devices which implement the converting of heat into electricity. But the traditional view often holds that thermoelectric effects in superconductors must be very weak, which mainly originates from the electron-hole symmetry of the SC density of states, thus giv-

ing zero thermoelectric response to a thermal gradient. Nevertheless, if one can break the particle-hole symmetry in terms of some methods, large thermoelectric effects can still be achieved. Thus, the violation of particle-hole symmetry can be realized in hybrid QD systems, where the advantage of QDs sandwiched between SC and metallic electrodes is that the manipulation of the electron-hole symmetry can readily be realized by an external gate voltage connected to the QDs. Indeed, lots of recent works have reported the remarkable thermoelectric properties in hybrid QD systems. In a normal-metal(N)-QD-SC system, it was found that the quasiparticle tunneling can contribute to the generation of high thermopower [31]. Intriguingly, a N[or ferromagnetic(FM)]-QD-SC system can be used as a high efficient thermoelectric diode [32]. Besides, the large charge(spin) thermopower can be obtained in a QD attached to FM and SC electrodes [33]. In a three-terminal hybrid structure based on double QDs, Hussein *et al.* showed the nonlocal electron-hole breaking of the system can generate nonlocal thermoelectric effects [34]. In particular, the thermopower of a FM-QD-SC system can be proposed as a probe to reveal the odd-frequency pairing in superconductors [35]. Remarkably, some recent investigations have suggested that hybrid QD systems can be operated as quantum heat engines or refrigerators based on different physical mechanisms [36, 37].

Note that a QD coupled to a molecular magnet forms a nanomagnet, and the magnetic moment of the nanomagnet will process about the an external magnetic field. Stadler *et al.* have elucidated the mechanism of the $0 - \pi$ transition in the nanomagnet [38]. In nanosystems analogous to the aforementioned nanomagnet, some interesting physical behaviors also occur, such as the spin pump [39], the Josephson diode [40], the conductance oscillation due to the spin-flip [41], etc. However, when a nanomagnet is put in the proximity of N and SC contacts, how will the spin-related thermoelectricity of this hybrid nanomagnet system exhibit interesting physics? To the best of our knowledge, this remains rarely explored. Accordingly, in this paper we will shed light on interesting thermoelectric properties involved in this hybrid system. Furthermore, some underlying physics is unraveled. Our results indicate the potential applications of the hybrid thermoelectric device based on spin caloritronic effects.

## II. PHYSICAL MODEL AND FORMULATION

The hybrid nanomagnet system under consideration is depicted in Fig. 1. It is comprised of the left N reservoir, a QD attached to a molecular magnet, and the SC reservoir. The total Hamiltonian describing the hybrid system reads $H = H_\nu + H_C + H_T$. The first term $H_\nu$ modeling the electrodes labeled by $\nu = N, SC$ is given as

$$H_\nu = \sum_{k\nu\sigma} \varepsilon_{\nu k} a^\dagger_{\nu k\sigma} a_{\nu k\sigma} + \delta_{\eta,SC} \sum_k (\Delta a^\dagger_{\nu k\uparrow} a^\dagger_{\nu-k-\downarrow} + H.c.), \tag{1}$$

where $\varepsilon_{\nu k}$ is the single particle energy of the electrode $\nu$, and $a^\dagger_{\nu k\sigma}(a_{\nu k\sigma})$ denotes the creation (annihilation) operator of an electron in the electrode $\nu$ with spin $\sigma = \uparrow, \downarrow$. $\Delta$ is the SC gap, which can be set as $\Delta > 0$ without loss of generality. The second term $H_C$, which describes the central region of the hybrid nanomagnet system consisting of a QD coupled to a molecular magnet, is written as

$$H_C = \sum_\sigma (\varepsilon_d + \sigma\Delta_z + \sigma J_s \cos\theta) d^\dagger_\sigma d_\sigma + \sum_\sigma J_s e^{-i\sigma\varphi} d^\dagger_\sigma d_{\bar\sigma}. \tag{2}$$

Here, $d^\dagger_\sigma(d_\sigma)$ is the the creation (annihilation) operator of the QD electron with the energy level $\varepsilon_d$ and spin $\sigma = \uparrow, \downarrow$. When the magnetic field $\vec{B} = (0, 0, B)$ is applied along the $z$ direction, the Zeeman energy $\Delta_z = \mu_B B$ as well as a precession of the spin of the molecular magnet will emerge. Here, $\mu_B$ denotes the Bohr magneton. In the following, we will explain other terms in Eq.(2). The magnetic field induces a torque $\vec{\mu} \times \vec{B}$ acting on the local magnetic moment $\vec{\mu}$ of the molecular magnet with $\vec{\mu} = -\gamma\vec{S}$. Here, $\gamma$ and $\vec{S}$ represent the gyromagnetic ratio and the spin of the molecular magnet, respectively. The equation of motion of the local magnetic moment is given as $\partial\vec{\mu}/\partial t = -\vec{\mu} \times \gamma\vec{B}$, thus we readily obtain the equation of the precession of the spin $\partial\vec{S}/\partial t = -\vec{S} \times \gamma\vec{B}$. The solution of this equation is given as $\vec{S} = (S\cos\varphi\sin\theta, S\sin\varphi\sin\theta, S\cos\theta)$. $\theta$ is the angle between $\vec{B}$ and $\vec{S}$, and $S$ is the magnitude of the spin. $\varphi = \omega_L t$ is the azimuthal angle with the Larmor frequency $\omega_L = \gamma B$. The exchange interaction $H_{SD}$ between the molecular magnet and the QD is given as

$$H_{SD} = \frac{\hbar}{2} \sum_\sigma W_s d^\dagger_\sigma (\vec{S} \cdot \vec{\sigma}) d_{\sigma'}, \tag{3}$$

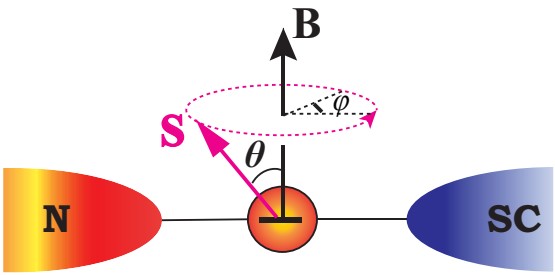

FIG. 1. Schematic illustration of a QD coupled to the spin $\mathbf{S}$ of a molecular magnet.

with the coupling strength $W_s$ between the spin and the QD. Here, $\vec{\sigma} = (\sigma_x, \sigma_y, \sigma_z)$ denotes the Pauli matrices. Actually, Eq.(3) can be further altered to

$$H_{SD} = \sum_\sigma \sigma J_s \cos\theta d_\sigma^\dagger d_\sigma + J_s e^{-i\varphi} d_\uparrow^\dagger d_\downarrow + J_s e^{i\varphi} d_\downarrow^\dagger d_\uparrow = \sum_\sigma \sigma J_s \cos\theta d_\sigma^\dagger d_\sigma + \sum_\sigma J_s e^{-i\sigma\varphi} d_\sigma^\dagger d_{\bar{\sigma}}, \quad (4)$$

with $J_s = \hbar W_s S/2$. Hence, we complete the description of the Hamiltonian $H_C$. Finally, $H_T$ describing the tunnel coupling between the QD and the external electrodes is characterized by

$$H_T = \sum_{\nu k\sigma} (t_\nu a_{\nu k\sigma}^\dagger d_\sigma + H.c.), \quad (5)$$

where $t_\nu$ represent the tunnel matrix elements.

Due to the presence of the SC electrode, it is very convenient for introducing the generalized Nambu basis $\Psi^\dagger = (d_\uparrow^\dagger, d_\downarrow, d_\downarrow^\dagger, d_\uparrow)$. By virtue of the nonequilibrium Green's function formalism, the spin-related charge current $J_\sigma$ and heat one $J_{Q\sigma}$, flowing from the N lead to

the SC one, can be determined by the formulas

$$J_\sigma = \frac{e}{h}\int d\varepsilon (f_N - \bar{f}_N)T_{A\sigma}(\varepsilon) + \frac{e}{h}\int d\varepsilon (f_N - f_S)T_{Q\sigma}(\varepsilon),$$

$$J_{Q\sigma} = \frac{e}{h}\int d\varepsilon (\varepsilon - \mu_N)(f_N - \bar{f}_N)T_{A\sigma}(\varepsilon) + \frac{e}{h}\int d\varepsilon (\varepsilon - \mu_N)(f_N - f_S)T_{A\sigma}(\varepsilon),$$

(6)

where $f_N = \{\exp[(\varepsilon - \mu_N)/k_B T_N] + 1\}^{-1}$ ($\bar{f}_N = \{\exp[(\varepsilon + \mu_N)/k_B T_N] + 1\}^{-1}$) stands for the Fermi-Dirac distribution of electrons (holes) in the N lead with the electrochemical potential $\mu_N$, the temperature $T_N$, and the Boltzmann constant $k_B$. $f_S = \{\exp[(\varepsilon - \mu_{SC})/k_B T_{SC}] + 1\}^{-1}$ denotes the Fermi-Dirac distribution of the SC lead with $T_S = T$ and $\mu_S = \mu = 0$. $T_{A\sigma}(\varepsilon)$ and $T_{Q\sigma}(\varepsilon)$ are the AR coefficient and the quasiparticle tunneling(QT) probability, respectively, which are explicitly expressed as $T_{A\uparrow}(\varepsilon) = (\Gamma^N)^2[|G_{12}^r(\varepsilon)|^2 + |G_{14}^r(\varepsilon)|^2]$, $T_{A\downarrow}(\varepsilon) = (\Gamma^N)^2[|G_{32}^r(\varepsilon)|^2 + |G_{34}^r(\varepsilon)|^2]$, $T_{Q\uparrow}(\varepsilon) = \Gamma^N\Gamma^{SC}\tilde{\rho}(\varepsilon)\{|G_{11}^r(\varepsilon)|^2| + G_{12}^r(\varepsilon)|^2 + |G_{13}^r(\varepsilon)|^2 + |G_{14}^r(\varepsilon)|^2 + 2\frac{\Delta}{\varepsilon}Re[G_{14}^r(\varepsilon)G_{13}^{r*}(\varepsilon) - G_{12}^r(\varepsilon)G_{11}^{r*}(\varepsilon)]\}$, and $T_{Q\downarrow}(\varepsilon) = \Gamma^N\Gamma^{SC}\tilde{\rho}(\varepsilon)\{|G_{31}^r(\varepsilon)|^2| + G_{32}^r(\varepsilon)|^2 + |G_{33}^r(\varepsilon)|^2 + |G_{34}^r(\varepsilon)|^2 + 2\frac{\Delta}{\varepsilon}Re[G_{34}^r(\varepsilon)G_{33}^{r*}(\varepsilon) - G_{32}^r(\varepsilon)G_{31}^{r*}(\varepsilon)]\}$. Here, $\tilde{\rho}(\varepsilon) = \theta(|\varepsilon| - \Delta)|\varepsilon|/\sqrt{\varepsilon^2 - \Delta^2}$. The detailed derivations of $T_{A\sigma}$ and $T_{Q\sigma}$ are given in the Supplemental Material.

We assume that along the hybrid nanomagnet system small temperature and chemical potential differences $\delta T = T_N - T$ and $\delta\mu = \mu_N - \mu = e\delta V$ are applied, where $\delta V$ is the bias voltage between the two leads. Within the linear response regime, the relationship between currents($J_\sigma$ and $J_{Q\sigma}$) and thermodynamic forces ($\delta V$ and $\delta T$) can be written as

$$\begin{pmatrix} J_\sigma \\ J_{Q\sigma} \end{pmatrix} = \begin{pmatrix} e^2 L_{11\sigma} & \frac{e}{T}L_{12\sigma} \\ eL_{21\sigma} & \frac{1}{T}L_{22\sigma} \end{pmatrix} \begin{pmatrix} \delta V \\ \delta T \end{pmatrix},$$

(7)

where $L_{mn\sigma}(m,n = 1,2)$ are the Onsager coefficients, which are explicitly given as

$$L_{11\sigma} = \frac{1}{h}\int d\varepsilon[2T_{A\sigma}(\varepsilon) + T_{Q\sigma}(\varepsilon)][-\frac{\partial f_0(\varepsilon)}{\partial\varepsilon}],$$

$$L_{12\sigma} = \frac{1}{h}\int d\varepsilon(\varepsilon - \mu)T_{Q\sigma}(\varepsilon)[-\frac{\partial f_0(\varepsilon)}{\partial\varepsilon}],$$

$$L_{21\sigma} = \frac{1}{h}\int d\varepsilon(\varepsilon - \mu)[T_{Q\sigma}(\varepsilon) + 2T_{A\sigma}(\varepsilon)][-\frac{\partial f_0(\varepsilon)}{\partial\varepsilon}],$$

$$L_{22\sigma} = \frac{1}{h}\int d\varepsilon(\varepsilon - \mu)^2 T_{Q\sigma}(\varepsilon)[-\frac{\partial f_0(\varepsilon)}{\partial\varepsilon}].$$

(8)

Here, $f_0(\varepsilon) = f(\varepsilon, \mu, T)$ is the equilibrium Fermi-Dirac function with the chemical potential $\mu$ and temperature $T$. Notice that $L_{12\sigma} \neq L_{21\sigma}$ arises from an applied magnetic field breaking time-reversal symmetry,

Next, we present thermoelectric coefficients which can unravel thermoelectric properties of the hybrid system. For $\delta T = 0$, the spin-related conductance can be calculated as $G_\sigma = J_\sigma/\delta V = e^2 L_{11\sigma}$. Thus, the charge (spin) conductance is given as $G_c = G_\uparrow + G_\downarrow$ ($G_s = G_\uparrow - G_\downarrow$). Under the vanishing charge current, the spin-related thermopower (Seebeck coefficient) is defined as

$$S_\sigma = \left(\frac{\delta V}{\delta T}\right)_{J_\sigma=0} = -\frac{1}{eT}\frac{L_{12\sigma}}{L_{11\sigma}}. \tag{9}$$

Similarly, the charge(spin) thermopower is expressed as $S_c = S_\uparrow + S_\downarrow$ ($S_s = S_\uparrow - S_\downarrow$). The heat conductance can be given as

$$\kappa_e = \sum_\sigma \left(\frac{J_{Q\sigma}}{\delta T}\right)_{J_\sigma=0} = \frac{1}{T}\sum_\sigma \left(L_{22\sigma} - \frac{L_{12\sigma}L_{21\sigma}}{L_{11\sigma}}\right). \tag{10}$$

At last, the charge(spin) figure of merit is given as $Z_c T = G_c S_c^2 T/(\kappa_e + \kappa_{ph})$ [$Z_s T = |G_s|S_c^2 T/(\kappa_e + \kappa_{ph})$]. Here, $\kappa_{ph}$ denotes the the heat conductance from the phonon contribution. This term, however, will be ignored, since we focus on the low temperature region where only electrons contribute effectively in heat transport. Thus, the above thermoelectric coefficients provide a useful tool for exploring spin thermoelectric properties of the hybrid nanomagnet setup.

## III. RESULTS AND DISCUSSION

In the following numerical analysis based on the above formulas, the SC gap $\Delta$ is taken as the energy unit. For simplicity, we take $\varphi = 0$, if not stated otherwise. Furthermore, we consider the symmetric coupling $\Gamma^N = \Gamma^{SC} = 0.1\Delta$ throughout our paper.

As is well known, the thermoelectric transport is a thermodynamic process. In order to quantitatively understand the thermodynamic nature of Seebeck effect involving charge carriers, we first illustrate the dependence of thermoelectric coefficients on the QD's energy level $\varepsilon_d$ for different temperatures $k_B T$, as shown in Fig. 2. At relatively low temperatures $k_B T = 0.1\Delta$, the conductance $G_c$ in Fig. 2(a) exhibits a narrow peak pinned at the Fermi level $\varepsilon_d = 0$, which mainly originates from the Andreev transport. As temperature increases, the central peak is reduced, but two-side peaks emerges outside the energy gap [red solid line in Fig. 2(a)]. This is because more quasiparticles participate in the thermoelectric transport. Since the heat conductance is mainly determined by the quasiparticle contribution, the heat conductance $\kappa_e$ at low temperatures displays the symmetric shape as well as two weak

peaks(black solid line) near the energy gap $\Delta$, as shown in Fig. 2(b). The increase of temperature induces the considerable enhancement of heat conductance, which implies that more quasiparticles are involved in the heat transport. As we know, the charge thermopower $S_c$ is connected with the generation of electrical voltage by a temperature gradient. As the the QD's energy level $\varepsilon_d$ varies, $S_c$ in Fig. 2(c) displays the antisymmetrical shape with respect to the Fermi level. Also, $S_c$ can change its signs by modulating $\varepsilon_d$. Notice that the positive(negative) $S_c$ corresponds to different types of carriers, namely, $S_c > 0(S_c < 0)$ means that the charge carriers are holes(electrons). When $k_B T$ is relatively small, the electron-hole symmetric region where the thermopower vanishes is relatively wide. However, the electron-hole symmetric region becomes narrow with the increase of temperature, indicating that the electron-hole symmetry is readily broken with increasing temperature [see the red solid line in Fig. 2(c)]. In a sense, the evolution of $S_c$ can be cast in the characteristic curves of charge figure of merit $Z_c T$, as displayed in Fig. 2(d). The two peaks of $Z_c T$ correspond to the peak and valley of $S_c$ in Fig. 2(c). Interestingly, one can see from Fig. 2(d) that the thermoelectric $Z_c T$ is pronouncedly enhanced and even far more than 1 at high temperature, which demonstrates the strong violation of the WF law.

Since the polar angle $\theta$ as an intrinsic and important quantity can influence the thermoelectric transport, we illustrate the evolution of thermoelectric coefficients for indicated values of $\theta$ in Fig. 3. For a static magnetic moment corresponding to $\theta = 0$, $S_c$ shows the alternate change of peaks and valleys, as presented in Fig. 3(a). With the presence of $\theta$, the peaks and valleys of $S_c$ shift towards the Fermi level, and their amplitudes gradually decrease. When the magnetic moment is precessing in the plane ($\theta = \pi/2$), $S_c$ almost vanishes in the roughly energy region (-2,2), which implies that the precession magnetic moment in the plane is not conductive to improving the charge thermopower. It is noteworthy that the spin thermopower $S_s$ is induced due to the significant imbalance of spin-up and spin-down channels of carriers[see dashed lines in Fig.3(a)]. Interestingly enough, a pure spin-Seebeck effect can be yielded, such as the point marked by the black arrow, where $S_s$ is nonzero with the vanishment of $S_c$. This is important because the pure spin-Seebeck effect can, in principle, serve as a generator of the pure spin current. Besides, the considerable spin thermopower $S_s$ can be achieved by means of tuning $\varepsilon_d$ and $\theta$. The realization of spin-Seebeck effect in our setup is irrespective of the magnetic electrodes compared to the previous works [33, 42], which constitutes the advantage of the hybrid device. The impact

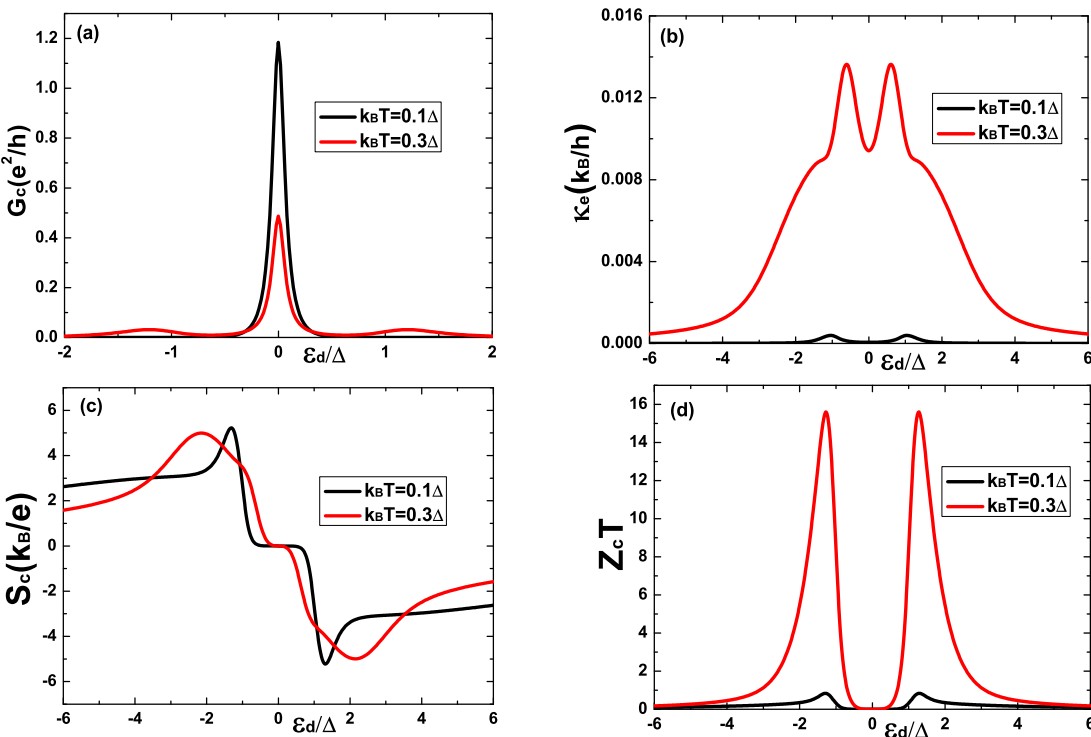

FIG. 2. Thermoelectric coefficients: (a) conductance $G_c$, (b) thermal conductance $\kappa_e$, (c) charge thmopower $S_c$, and (d) charge figure of merit $Z_cT$ as a function of the QD's energy level $\varepsilon_d$ for indicated values of the temperature $k_BT$. Other parameters are taken as $\theta = \pi/4$, $\Delta_z = 0.05\Delta$, and $J_s = 0.05\Delta$.

of $\theta$ on the charge and spin thermoelectric efficiency($Z_cT$ and $Z_sT$) are depicted in Figs.3(b) and 3(c). As $\theta$ increases, $Z_cT$ and $Z_sT$ exhibit the symmetric line shapes with regard to the Fermi level, but the dependence of $Z_sT$ on $\theta$ is of more significant compared to $Z_cT$. Not only can one observe the enhanced thermoelectric efficiency as shown in Fig. 3(b), but also the spin thermoelectric efficiency $Z_sT$ is close to 1 or even more than 1, as described in Fig. 3(c). The remarkable spin thermoelectric efficiency is beneficial to the design of spin caloritronic devices. Indeed, the violation of the WF law generally appear in nanosacle systems. The Lorenz ratio, defined by $L = \kappa_e/G_cT$ in units of $L_0 = \pi^2 k_B^2/3e^2$, can characterize the breakdown of the WF law. Fig. 3(d) reveals the features of the Lorenz ratio $L/L_0$ for specific values of $\theta$. Although $L/L_0$ shows a certain distinction for different $\theta$, $L/L_0$ deviates from 1 in most region of $\varepsilon_d$, indicating the remarkable violation of the universal WF law.

In reality, the exchange energy $J_s$ between the local spin and the QD, which is equivalent

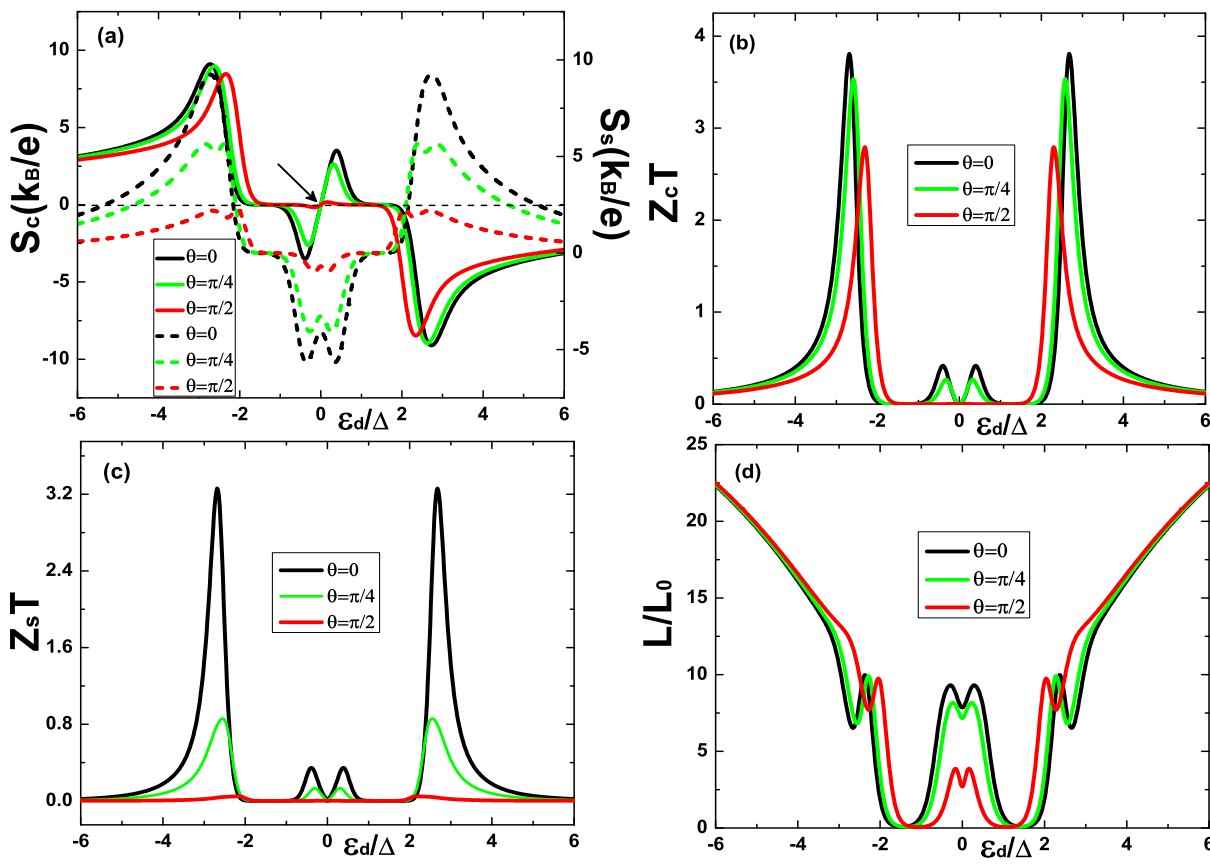

FIG. 3. Thermoelectric coefficients: (a) charge (spin) thermopower $S_c$ ($S_s$), (b) charge figure of merit $Z_cT$, (c) spin figure of merit $Z_sT$, and (d) Lorenz ratio $L/L_0$ as a function of the QD's energy level $\varepsilon_d$ for indicated values of the polar angle $\theta$. Other parameters are taken as $k_BT = 0.1\Delta$, $\Delta_z = 0.5\Delta$, and $J_s = 1.0\Delta$.

to a spin-flip mechanism, can significantly influence the thermoelectric features of the hybrid nanomagnet. Color maps of thermopowers ($S_c$ and $S_s$) as a function $J_s$ and $\varepsilon_d$ are plotted in Figs. 4(a1) and 4(a2). Clearly, $S_c(S_s)$ shows the antisymmetrical (symmetric) structure with respect to the lines $\varepsilon_d = 0$. It is interesting to note that there exist more resonances corresponding to peaks and valleys with the increase of $J_s$ as shown in Figs. 4(a1)-4(a3). Physically, this is because the strong $J_s$ can lead to the splitting of the QD's energy level, and thus more resonant tunneling channels are involved in the thermoelectric transport. It

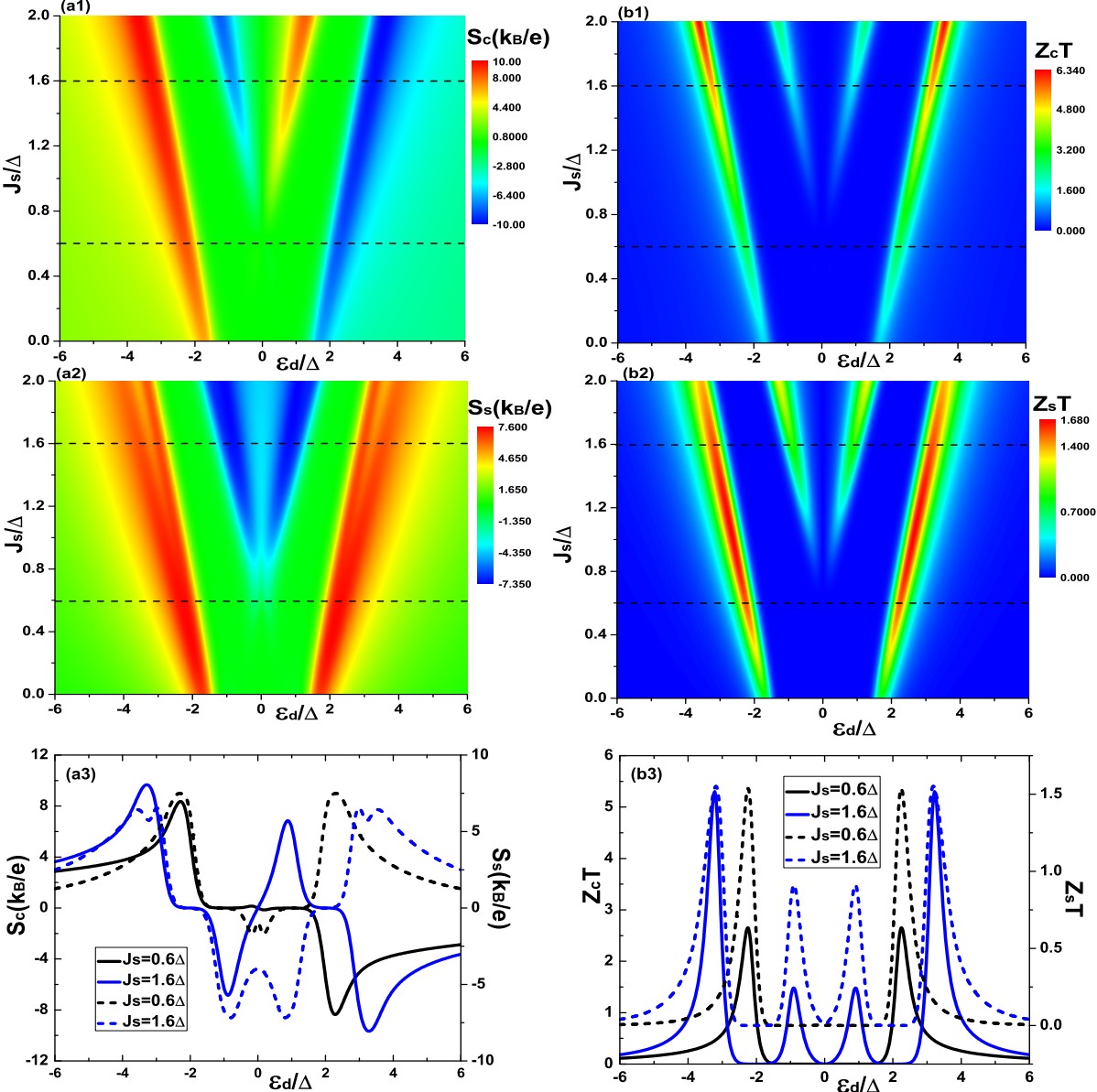

FIG. 4. In the left column: charge (spin) thermopower $S_c$ ($S_s$) as a function of the coupling energy $J_s$ and the QD's energy level $\varepsilon_d$ is shown in (a1)[(a2)]. (a3) The relevant cross sections of $S_c$ (solid lines) and $S_s$ (dashed lines) for indicated values of $J_s$, and these cross sections are indicated by dashes lines in (a1) and (a2). In the right column: charge (spin) figure of merit $Z_cT$ ($Z_sT$) as a function of the coupling energy $J_s$ and the QD's energy level $\varepsilon_d$ is shown in (b1)[(b2)]. (b3) The relevant cross sections of $Z_cT$ (solid lines) and $Z_sT$ (dashed lines) for indicated values of $J_s$, and these cross sections are indicated by dashes lines in (b1) and (b2). Other parameters are taken as $k_BT = 0.1\Delta$, $\theta = \pi/6$, and $\Delta_z = 0.5\Delta$.

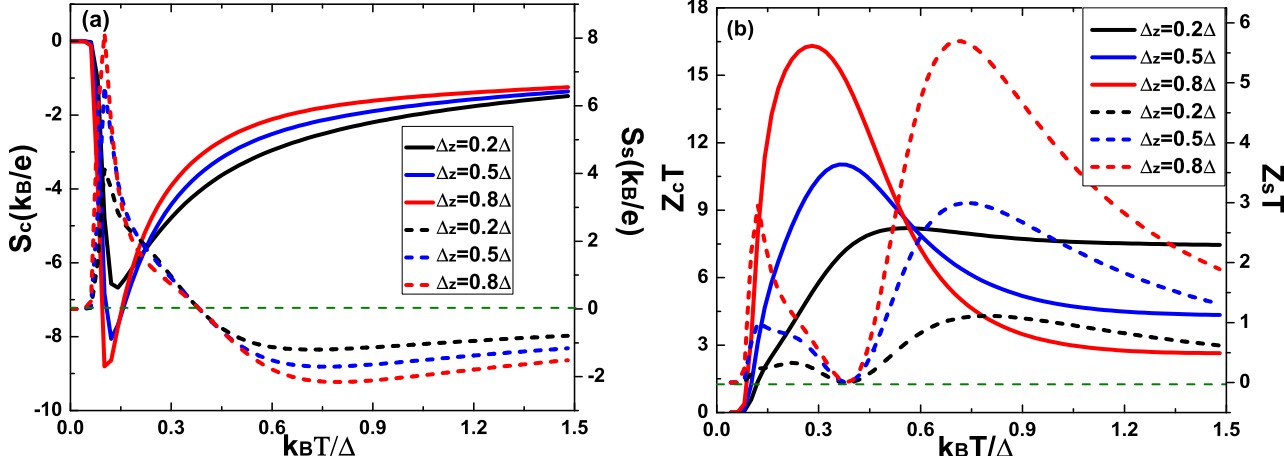

FIG. 5. (a) Charge (spin) thermopower $S_c$ ($S_s$) as a function of the temperature $k_B T$ for indicated values of the Zeemann energy $\Delta_z$. (b) Charge (spin) figure of merit $Z_c T$ ($Z_s T$) as a function of the temperature $k_B T$ for indicated values of the Zeemann energy $\Delta_z$. Other parameters are taken as $\varepsilon_d = 2.5\Delta$, $\theta = \pi/6$, and $J_s = 0.5\Delta$.

is expected that the strong $J_s$ is more likely to induce the emergence of spin-Seebeck effect as well as the generation of the considerable pure spin-Seebeck effect[see the blue dashed line in Fig. 4(a3)]. In comparison with the contour plots of $S_c$ and $S_s$, both $Z_c T$ and $Z_s T$ display the visible symmetric about the lines $\varepsilon_d = 0$[see Figs. 4(b1) and 4(b2)]. Indeed, this can be qualitatively understood from the expressions of $Z_c T$ and $Z_s T$. For the charge thermoelectric efficiency $Z_c T$, increasing $J_s$ not only causes the appearance of more peaks in $Z_c T$ plots, but also the magnitude of $Z_c T$ is enhanced[see solid lines in Fig. 4(b3)]. As can be seen, $Z_s T$ exhibits similar characteristics to $Z_c T$ as $J_s$ grows. Especially, $Z_s T$, which determines the spin thermoelectric efficiency of the hybrid thermoelectric system, can exceed 1 [see the blue dashed line in Fig. 4(b3)]. Actually, the enhancement of $Z_s T$ attributes to the increase of $J_s$, which is of significance for developing high-performance spin-thermoelectric devices.

Finally, we discuss the $k_B T$ dependence of thermopowers and figures of merit for indicated values of the Zeemann energy $\Delta_z$, as shown in Fig. 5. In Fig. 5(a), One can clearly see that the charge $S_c$ and the spin $S_s$ thermopowers display the nonmonotonic evolution with increasing temperature. At fairly low temperatures, $S_c$ and $S_s$ almost vanish. As a matter of fact, thermopowers mainly arise from the contribution of quasiparticles outside the energy gap. When the system is at very low temperatures, the excitation of quasiparticles is almost entirely suppressed, which leads to the disappearance of $S_c$ and $S_s$. As temperature increases, $|S_c|$ and $S_s$ first increase, and then decrease after attaining optimal values. Such nonmonotonic behavior can be understood as follows. On the one side, more and more carriers are involved in thermoelectric transport due to the increase of temperature, which, in principle, gives rise to the enhancement of $|S_c|$ and $S_s$. On the other side, the amplitude of $-\partial f_0 / \partial \varepsilon$ becomes weak with increasing temperature (see the expression of $L_{12\sigma}$). Therefore, the competition between the above two factors induce the optimal thermopowers, as described in Fig. 5(a). One also notes that increasing the Zeemann energy $\Delta_z$ can enhance thermopowers as well as thermoelectric efficiency in such nanostructure, which is due to the fact that the large $\Delta_z$ makes the energy level free from the binding of the SC gap, and thus more quasiparticles contribute to the enhancement of thermopowers. In particular, the spin thermopower $S_s$ changes its sign as temperature grows, indicating that one can control the direction of thermally driven spin currents by tuning the temperature. Indeed, the extreme values of thermopowers, to some extent, can be reflected in figures of merit, as shown in Fig. 5(b). Again, one observes that $Z_s T$ can surpass 1 at relatively high temperatures [see red and blue dashed lines in Fig. 5(b)], which is important for practical applications in spin thermoelectric devices.

## IV. CONCLUSIONS

In summary, a hybrid thermoelectric device consisting of a nanomagnet coupling to N and SC leads is put forward in order to deeply understand the spin-related thermoelectric transport. In terms of the nonequilibrium Green's technique, we have analyzed numerically thermoelectric properties of this hybrid nanomagnet system. The increase of temperature can change the shapes and magnitudes of the charge(heat) conductance $G_c$ ($k_e$). Since the QD's energy level is controlled by gate voltage, it is easy to change the sign of the

thermopower $S_c$. Furthermore, the charge figure of merit $Z_c T$ is more than 1 at high temperatures, indicating that the validity of the WF law is broken. Thermoelectric quantities exhibit the obvious $\theta$ dependence. Importantly, a pure spin-Seebeck effect can be produced with the vanishment of $S_c$, which implies that the hybrid nanomagnet can in practice act as a pure spin-current generator. Besides, the large $Z_s T$ and $L/L_0$ demonstrate the clear advantages of heat-to-electrical energy conversion for this hybrid thermoelectric device. In the parameter space $(J_s, \varepsilon_d)$, one can modulate the shapes and amplitudes of thermopowers ($S_c$ and $S_s$) and figures of merit($Z_c T$ and $Z_s T$), and thus it is possible to obtain the considerable spin thermoelectric conversion. $|S_c|(S_s)$ and $Z_c T(Z_s T)$ increase with temperature, and exhibit the optimal values in respective curves. In fact, the spin thermopower $S_s$ reflects thermoelectric characteristics mediated by electron spins, which can serve as a sensitive probe for spin properties of molecular magnets. One finds that $S_s$ can change its sign with modulating the temperature, indicating that the direction of thermally excited spin currents can be manipulated. Furthermore, the large spin thermoelectric efficiency can be achieved by varying the temperature. Our results provide useful insights into understanding and engineering spin caloritronic devices.

## ACKNOWLEDGMENTS

This work is financially supported by the Fundamental Research Funds for the Central Universities (Grant No. 2020ZDPYMS32).

## Appendix A: The explicit expressions of AR and QT transmission coefficients

In terms of the non-equilibrium Green's function method, here we present the explicit derivations of the AR coefficient and the QT probability, which is omitted in the main text.

The charge current through the normal-metal(N) lead is written as [42, 43]

$$J_{\uparrow(\downarrow)} = \frac{ie}{\hbar} \int \frac{d\varepsilon}{2\pi} Tr\{\hat{\sigma}_z \Gamma_N [G^<(\varepsilon) + F_N(G^r(\varepsilon) - G^a(\varepsilon))]\}_{11(33)}, \qquad (A1)$$

where $\hat{\sigma}_z$ is a $4 \times 4$ matrix where the Pauli matrix $\sigma_z$ is its diagonal components. Because of the presence of the superconducting(SC) lead, we introduce the Nambu basis $\Psi^\dagger = (d_\uparrow^\dagger, d_\downarrow, d_\downarrow^\dagger, d_\uparrow)$ in order to investigate the Andreev transport regime. Via the Dyson

equation, the retarded Green's function $G^r(\varepsilon)$ of the total hybrid structure is

$$[G^r(\varepsilon)]^{-1} = [g^r(\varepsilon)]^{-1} - \Sigma^r_N - \Sigma^r_{SC}, \tag{A2}$$

where $g^r(\varepsilon)$ represents the retarded Green's function of the nanomagnet isolated from the leads, which is given as

$$[g^r(\varepsilon)]^{-1} =$$

$$\begin{pmatrix} \varepsilon - \varepsilon_d - \Delta_z - J_s\cos\theta & 0 & -J_s e^{-i\varphi} & 0 \\ 0 & \varepsilon + \varepsilon_d - \Delta_z - J_s\cos\theta & 0 & J_s e^{-i\varphi} \\ -J_s e^{i\varphi} & 0 & \varepsilon - \varepsilon_d + \Delta_z + J_s\cos\theta & 0 \\ 0 & J_s e^{i\varphi} & 0 & \varepsilon + \varepsilon_d + \Delta_z + J_s\cos\theta \end{pmatrix}. \tag{A3}$$

$\Sigma^r_N(\Sigma^r_{SC})$ in Eq.(A2) is the retarded self-energy due to the nanomagnet and the N(SC) lead. In the wide-band approximation, $\Sigma^r_N$ can be written as

$$\Sigma^r_N = -\frac{i}{2}\Gamma_N = -\frac{i}{2}\begin{pmatrix} \Gamma^N & 0 & 0 & 0 \\ 0 & \Gamma^N & 0 & 0 \\ 0 & 0 & \Gamma^N & 0 \\ 0 & 0 & 0 & \Gamma^N \end{pmatrix}, \tag{A4}$$

where $\Gamma^N$ is the coupling strength defined by $\Gamma^N = 2\pi\sum_k |t_N|^2\delta(\varepsilon - \varepsilon_{Nk})$. On the other hand, $\Sigma^r_{SC}$ can be given as

$$\Sigma^r_N = -\frac{i}{2}\Gamma_{SC} = -\frac{i}{2}\rho(\varepsilon)\Gamma^{SC}\begin{pmatrix} 1 & -\frac{\Delta}{\varepsilon} & 0 & 0 \\ -\frac{\Delta}{\varepsilon} & 1 & 0 & 0 \\ 0 & 0 & 1 & \frac{\Delta}{\varepsilon} \\ 0 & 0 & \frac{\Delta}{\varepsilon} & 1 \end{pmatrix}, \tag{A5}$$

where $\rho(\varepsilon)$ denotes a dimensionless modified BCS density of states of the SC lead, which is defined by

$$\rho(\varepsilon) = \frac{\varepsilon\theta(\Delta - |\varepsilon|)}{i\sqrt{\Delta^2 - \varepsilon^2}} + \frac{|\varepsilon|\theta(|\varepsilon| - \Delta)}{\sqrt{\varepsilon^2 - \Delta^2}}. \tag{A6}$$

$\Gamma^{SC} = 2\pi|t_{SC}|^2 N_{SC}$ in Eq.(A5) denotes the coupling of the SC lead to the nanomagnet, where $N_{SC}$ is the density of states in the normal state of the SC lead. It is worth noting that the coupling matrices $\Gamma_{\nu=N,SC} = i(\Sigma^r_\nu - \Sigma^a_\nu)$ with $\Sigma^r_\nu = [\Sigma^a_\nu]^\dagger$. By means of the Keldysh

relation, the lesser Green's function $G^<(\varepsilon)$ is written as $G^<(\varepsilon) = G^r(\varepsilon)(\Sigma_N^< + \Sigma_{SC}^<)G^a(\varepsilon) = G^r(\varepsilon)(iF_N\Gamma_N + if_S\Gamma_{SC})G^a(\varepsilon)$ with the advanced Green's function $G^a(\varepsilon) = [G^r(\varepsilon)]^\dagger$. In Eq.(1), $F_N$ is the matrix of Fermi-Dirac functions for the N lead and takes the form

$$F_N = \begin{pmatrix} f_N & 0 & 0 & 0 \\ 0 & \bar{f}_N & 0 & 0 \\ 0 & 0 & f_N & 0 \\ 0 & 0 & 0 & \bar{f}_N \end{pmatrix}. \tag{A7}$$

Besides, we have the following identity

$$G^r(\varepsilon) - G^a(\varepsilon) = -iG^r(\varepsilon)(\Gamma_N + \Gamma_{SC})G^a(\varepsilon). \tag{A8}$$

By combing the above formulas with Eq.(A1), $T_{A\sigma}(\varepsilon)$ and $T_{Q\sigma}(\varepsilon)$ after some algebraic manipulations are obtained as

$$
\begin{aligned}
&T_{A\uparrow}(\varepsilon) = (\Gamma^N)^2[|G_{12}^r(\varepsilon)|^2 + |G_{14}^r(\varepsilon)|^2], \quad T_{A\downarrow}(\varepsilon) = (\Gamma^N)^2[|G_{32}^r(\varepsilon)|^2 + |G_{34}^r(\varepsilon)|^2], \\
&T_{Q\uparrow}(\varepsilon) = \Gamma^N\Gamma^{SC}\tilde{\rho}(\varepsilon)\{|G_{11}^r(\varepsilon)|^2| + G_{12}^r(\varepsilon)|^2 + |G_{13}^r(\varepsilon)|^2 + |G_{14}^r(\varepsilon)|^2 \\
&+ 2\frac{\Delta}{\varepsilon}Re[G_{14}^r(\varepsilon)G_{13}^{r*}(\varepsilon) - G_{12}^r(\varepsilon)G_{11}^{r*}(\varepsilon)]\}, \\
&T_{Q\downarrow}(\varepsilon) = \Gamma^N\Gamma^{SC}\tilde{\rho}(\varepsilon)\{|G_{31}^r(\varepsilon)|^2| + G_{32}^r(\varepsilon)|^2 + |G_{33}^r(\varepsilon)|^2 + |G_{34}^r(\varepsilon)|^2 \\
&+ 2\frac{\Delta}{\varepsilon}Re[G_{34}^r(\varepsilon)G_{33}^{r*}(\varepsilon) - G_{32}^r(\varepsilon)G_{31}^{r*}(\varepsilon)]\},
\end{aligned} \tag{A9}
$$

with $\tilde{\rho}(\varepsilon) = \theta(|\varepsilon|-\Delta)|\varepsilon|/\sqrt{\varepsilon^2 - \Delta^2}$. Thus, we achieve the AR and QT probabilities presented in the main text.

---

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
