# Peer review of "Spin caloritronics in a hybrid nanomagnet"

_SciPost Physics_

## Round 1 · Referee Report · Anonymous (Referee 1) · 2026-1-2

Strengths

1 - thermoelectric transport is an interesting topic 2- superconductor-nanomagnet hybrids as platform for quantum transport

Weaknesses

1- possible mistake in the calculation 2- unclear assumptions 3- small mistakes in formulas 4-physical realisation remains unclear

Report

Bai and coworkers investigate a hybrid system consisting of a quantum dot coupled to a nanomagnet between a normal metal and a superconductor. Using a nonequilibrium Green's function approach they find the coefficients for Andreev reflection and quasiparticle transport. From that they determine the thermoelectric response matrix and extract key quantities like the figure of merit .

The topic is of interest for communities working on superconducting transport and/or quantum thermodynamics. The research is therefore timely and will certainly find a readership. But if the impact is strong enough for SciPost Physics is not clear at the present stage.

There are, however, several issues that need to be resolved before the manuscript should get further consideration.

Critical issues are 1) The main Hamiltonian contains a magnetic field that leads to a Zeeman splitting of the dot level and to a precession of the nanomagnet (seen in the time dependent phase phi(t)). Now, on one hand a magnetic field would also act in the leads, but it is neglected here. Second, the time-dependent precession cannot be treated by simply going to energy-space. In the manuscript the phase is simply set to zero, which is not a solution of the equations. The authors need to clarify this inconsistency. 2) What is the actual Hamiltonian anyway? How is H_SD related to H_C. One can guess its, but it is not written 3) In the introduction, it is stated that the Wiedemann-Franz law eads to ZT<1. This is incorrect. The WF law relates electric and thermal conductance, but not the Seebeck coefficient. 4) Eq. (6) second line: The second integral contains T_A, which seems to make no sense. 5) After Eq. (10) ZT_c and ZT_s are defined. What is the motivation behind ZT_s? Is it related to some measurable quantity? 6) T_A should be spin-independent and symmetric in energy. This should simplify certain expressions. E.g. it should vanish from L_21. 7) In several figures (3a, 4a3, 4b3, 5) there are dashed and solid lines and two scales, but it is unclear which line belongs to which quantity.

This is a first list of points that need to be clarified, before the manuscript should be given further consideration. In its present form it is not publishable.

The results are interesting but the relation to experiments is unclear. Hence the relevance needs to be improved. The present version after careful consideration of the above points seems more suitable to be considered for SciPost Physics Core

Requested changes

See report.

Recommendation

Ask for major revision

  • validity: low
  • significance: low
  • originality: good
  • clarity: low
  • formatting: below threshold
  • grammar: below threshold

---

## Round 1 · Referee Report · Anonymous (Referee 2) · 2026-1-5

Strengths

Treats a quantum dot device in a new configuration

Weaknesses

Surprising unfamiliarity with ewperimental constraints

Report

“Spin caloritronics in a hybrid nanomagnet” refers to a theoretical study of “a thermoelectric device based on a hybrid nanomagnet”. This “device”, sketched in Figure 1, is a quantum dot at the intersection of a metal and a superconductor subject to a magnetic field. No attempt has been made to clarify the location of electrodes or thermometers on this “device”. Nevertheless, according to the abstract, it investigates “the evolution characteristics of thermoelectric coefficients in physically allowed parameter regions” and it finds that “not only is the Wiedemann-Franz strongly violated, but also a pure spin Seebeck effect” and “a considerable spin thermoelectric efficiency.”
As an experimentalist, I cannot verify the calculations, but because of the level of unfamiliarity with experimental facts, I cannot recommend the publication of this paper in any scientific journal.
Here is a list of obvious fallacies:
-Wiedemann-Franz law does not hold in superconductors. Therefore, speaking of its “violation” is surprising.
-Given that any such device will be mounted on a substrate, what matters in setting the thermoelectric figure of merit is not the thermal conductivity of electrons in the device, but the thermal conductivity of phonons in the substrate.
-Contrary to what is plotted in Fig. 2b, k_B^2/ \hbar (in W/K^2) are the natural units of the electronic thermal conductivity DIVIDED by temperature.
Submitted as a mathematical exercise, the paper would have been harmless. On the other hand, presented in the context of the search for “thermoelectric devices which implement the converting of heat into electricity”, it misleads its potential readers.

Recommendation

Reject

---

## Editorial Decision

in_refereeing